# Moisture Distribution and Structural Properties of Frozen Cooked Noodles with NaCl and Kansui

**DOI:** 10.3390/foods10123132

**Published:** 2021-12-17

**Authors:** Jiarong Wang, Yangyue Ding, Mingyang Wang, Tianqi Cui, Zeyu Peng, Jianjun Cheng

**Affiliations:** Grain Engineering Research Laboratory (10), College of Food Science, Northeast Agricultural University, Harbin 150030, China; wjrsci@163.com (J.W.); dingyangyue77@163.com (Y.D.); wmy15663028822@163.com (M.W.); cuitianqi0906@163.com (T.C.); zerypaul@163.com (Z.P.)

**Keywords:** frozen cooked noodles, NaCl, kansui, moisture distribution, structure, rheology, quality

## Abstract

The effects of NaCl (1–3%) and kansui (0.5–1.5%) on the quality of frozen cooked noodles (FCNs) were investigated, which provided a reference for alleviating the quality deterioration of FCNs. Textural testing illustrated that the optimal tensile properties were observed in 2% NaCl (N-2) and the maximum hardness and chewiness were reached at 1% kansui (K-1). Compared to NaCl, the water absorption and cooking loss of recooked FCNs increased significantly with increasing kansui levels (*p* < 0.05). Rheological results confirmed NaCl and kansui improved the resistance to deformation and recovery ability of thawed dough; K-1 especially had the highest dough strength. SEM showed N-2 induced a more elongated fibrous protein network that contributed to the extensibility, while excessive levels of kansui formed a deformed membrane-like gluten network that increased the solid loss. Moisture analysis revealed that N-2 reduced the free water content, while K-1 had the lowest freezable water content and highest binding capacity for deeply adsorbed water. The N-2 and K-1 induced more ordered protein secondary structures with stronger intermolecular disulfide bonds, which were maximally improved in K-1. This study provides more comprehensive theories for the strengthening effect of NaCl and kansui on FCNs quality.

## 1. Introduction

Frozen cooked noodles (FCNs), as a new type of instant noodle product, have grown increasingly popular among consumers due to their advantages in preservation, safety, and convenience, and now account for approximately 45% of the total noodle production [1,2]. However, freezing might reduce hardness, tensile force, aroma, and consumer acceptance of noodles [3]. These negative changes are closely related to the uneven water redistribution in the noodle matrix upon freezing and ice recrystallization caused by temperature fluctuations, which could lead to the loss of gluten network integrity and denseness, with implications for the machinability and rheological properties of the dough [4]. Currently, numerous studies emphasized the minimization of freeze-induced structural damage and quality deterioration during FCNs freezing.

Salts have been widely used as dough conditioners or quality improvers to improve the flavor, texture, and cooking characteristics of noodles. Depending on the types of salt in the recipe, the noodles are divided into two different types: white salted noodles containing NaCl and yellow alkaline noodles with Na_2_CO_3_ or K_2_CO_3_, or a mixture of them, called kansui [5]. Fu (2008) reported that the typical addition of salt was 1–3% by flour weight, while approximately 0.5–1.5% of alkaline salts was used to make alkaline noodles [6]. NaCl could increase the mixing tolerance of the dough, extend the dough development time, and improve the dough sheeting properties and noodle quality by increasing the density of gluten network [7,8,9]. The alkaline salt contributed to the unique color, texture, and flavor of the noodles and promoted a denser gluten structure by strengthening the disulfide bonds and non-disulfide bonds cross-linking [10,11,12]. In spite of the addition of NaCl and kansui being a convenient and cost-effective means of improving the noodles quality, few studies have investigated their effects on frozen cooked noodles.

The deterioration process of FCNs quality is strongly dependent on the quantity, physical state, and location of the water [2]. Previous studies demonstrated that the sodium and chloride ions in the NaCl are dispersed around the proteins and act to retain the water. NaCl also accelerates the absorption of water into the flour and distributes it uniformly [8]. In addition, Li et al. (2017) revealed that alkali enhanced the interaction between aqueous and non-aqueous components [11]. However, most studies about NaCl and kansui focused on the interactions between them and fresh dough [8,13]. Few quantitative studies have focused on the effects of NaCl and kansui on frozen noodles, and even fewer studies have taken the water molecules movement into account.

The singularity of this work was to investigate the effect of different levels of NaCl (1–3%) and kansui (0.5–1.5%) on the quality of FCNs, and to elucidate the potential mechanisms for the transformation of noodle quality in terms of rheological behavior, microscopic morphology, water distribution, and protein structure of the dough. This study might provide a more comprehensive theoretical basis for broadening the applications of NaCl and kansui in the noodle industry.

## 2. Materials and Methods

### 2.1. Materials

Wheat flour (9.80% moisture, 0.67% ash, and 11.75% protein) was obtained from Jinshahe Flour Industry Co., Ltd. (Xingtai, Hebei, China). Sodium chloride, sodium carbonate, and potassium carbonate were provided by Tianli Chemical Reagent Co., Ltd. (Tianjin, China). All reagents were analytical grade.

### 2.2. Preparation and Sampling of FCNs

The formulation of FCNs was composed of 200 g wheat flour, 70 g distilled water, and different levels of NaCl or kansui (%, *w*/*w*). The predetermined amount of NaCl or kansui was dissolved in water before adding. NaCl was added at 1%, 2%, and 3% levels; kansui (a 9:1 mixture of sodium and potassium carbonate) was added at 0.5%, 1%, and 1.5% levels; and the samples were named N-1, N-2, N-3, K-0.5, K-1, and K-1.5, respectively. The noodle without NaCl or kansui was the control.

The crumbly dough was obtained using an automatic dough mixer (M6-L30, Joyoung, Co., Ltd., Beijing, China) to knead for 5 min. The dough was rested at 23 °C for 30 min. Subsequently, the dough was placed on a semiautomatic sheeting machine (Juxin Machinery Co., Ltd., Zaoyang, China) and the sheeting gaps were reduced to 1 mm gradually to obtain dough sheets. The resultant dough sheets were cut into noodle strands (2 mm in width and 1 mm in thickness) with a cutter. Fresh noodles (30 g/500 mL water) were boiled for 3.5 min (optimal cooking time), and then immersed in 500 mL cold water (4 °C) for 1 min. These noodles were drained for 1 min and placed in sealed bags. Then, they were frozen in an ultra-low temperature freezer (DW-HL100, MELNG, Hefei, China), with air temperature in convection at −40 °C, until the core temperature dropped to −18 °C (~60 min). After freezing, the FCNs were stored at −18 ± 2 °C in the freezer (BCD-480WDGB, Haier Co., Ltd., Qingdao, China) for more than 24 h until use. Additionally, some fresh dough sheets were cut into 35 mm diameter discs and placed in polyethylene bags and frozen in the same procedure for further rheology testing [14].

### 2.3. Rheological Properties Tests

After fixed-time frozen storage, the dough sheets were removed from the polyethylene bags and put in a stainless-steel tray. Then, they were thawed in a constant temperature incubator at 25 °C and 75% relative humidity for 1 h. Then, the rheology performance of doughs (~3 g) was determined by a Haake Mars 40 rheometer (Thermo Fisher Scientific, Waltham, MA, USA) following the method of Zhang et al. (2018) and equipped with a 35 mm parallel plate geometry at a gap of 1 mm [15].

#### 2.3.1. Dynamic Frequency Sweep

A frequency sweep was run from 0.1–100 rad/s at a strain of 0.2% (within the linear viscoelastic region) and a temperature of 25 ± 0.1 °C to determine the storage modulus (G′), loss modulus (G′′) and tan δ (G′′/G′) as functions of frequency [16]. The degree of dependence of G′ on the frequency sweep (z′) and the strength of the dough (K) was obtained by fitting the curves to the following the power-law model (Equation (1)):G′ = K ω^z′^,(1)
where ω is the angular frequency.

#### 2.3.2. Creep Recovery Measurement

At 25 °C, within the linear viscoelastic range of the doughs (0.2% strain), instant stress (250 Pa) was applied and maintained for 300 s, and then stress was released to allow sample recovery for the next 300 s. The experimental parameters were obtained by analyzing the creep and recovery curves using Haake RheoWin 4.83.0004 software (Thermo Fisher Scientific, Waltham, MA, USA), including maximum creep compliance (J_max_), zero shear viscosity (η_0_), relative elastic part of the maximum creep compliance (J_e_/J_max_), and the relative viscous part of the maximum creep compliance (J_v_/J_max_).

### 2.4. Scanning Electron Microscope (SEM)

The cross-sections of the frozen-dried FCNs were examined using a scanning electron microscope (Hitachi S-3400N) at 500× magnification. The accelerating voltage for scanning was 5 kV [17].

### 2.5. Freezable Water Content

The freezable water (FW) content was measured by differential scanning calorimeter (DSC 3, Mettler Toledo, Basel, Switzerland) referring to the method described by Hong et al. (2021) [14]. A sliced subsample of ~10 mg from each sample was placed into the DSC pan. Then, the pan was hermetically sealed. During the test, samples were held at −40 °C for 10 min for the equilibrium of temperature and then heated from −40 °C to 40 °C at 10 °C/min. Nitrogen was used as a carrier gas at a flow rate of 20 mL/min. The FW content (FW%) was calculated by the following formula.
(2)$$\mathrm{FW}\%=\frac{\Delta H_{w}}{\Delta H_{i}\times \mathrm{TW}}\times 100\%$$

ΔH_w_, melting enthalpy, J/g of water. ΔH_i_, the latent heat of the ice fusion, 334.3 J/g. TW, the moisture content of the sample, *g*/*g*.

### 2.6. Measurement of Protons Migration and Distribution

The water distribution was determined using a 23-MHz ^1^H low-field nuclear magnetic resonance analyzer (LF-NMR, MesoMR23-060H-I, Suzhou Niumag Electronic Technology Co., Ltd., Suzhou, China) equipped with standard micro imaging accessories. The recooked FCNs (2.50 g) were weighed accurately and sealed in PET/PE bags to avoid the interference of air and moisture [4].

For the LF-NMR test, the transverse relaxation curves were acquired using the Carr–Purcell–Meiboom–Gill pulse sequence. The parameters employed are as follows: echo time (TE) = 0.2 ms, the number of echoes (NECH) = 10,000, the interval time of sampling (TW) = 2500 ms, and the number of scans (NS) = 2.

The water migration of FCNs was observed using magnetic resonance imaging (MRI). A 15-mm radiofrequency (RF) coil was selected and a standard SPIN-ECHO (SE) sequence was used to produce images. The parameters were: TE = 20 ms, TR = 500 ms, and Averages = 3.

### 2.7. Attenuated Total Reflectance-Fourier Transform Infrared Spectroscopy (ATR-FTIR) Analysis

The molecular interactions of freeze-dried dough were analyzed using ATR-FTIR (Nicolet iS50, Thermo Fisher, Waltham, MA, USA) according to the method of Ying et al. (2019) [18]. The spectra were generated in absorption mode in mid-IR (ca. 4000–525 cm^−1^) at a resolution of 4 cm^−1^ in 64 scans. The positions of the absorbance peaks located in the amide I region were determined using Fourier self-deconvolution and the second derivative using Peak Fit software (version 4.12). The peaks in the range 1650–1660 cm^−1^ corresponded to α-helix, 1660–1700 cm^−1^ and 1610–1640 cm^−1^ represented β-turn and β-sheet, respectively, and 1640–1650 cm^−1^ indicated the random coil.

### 2.8. Measurement of Free Sulfhydryl Content

The content of the free sulfhydryl (-SH) was measured as described by Zhang et al. (2018) using a UV-visible spectrophotometer (UVmini-1240, Shimadzu Instruments Co., Ltd., Suzhou, China) [15]. The -SH content was calculated as follows (Equation (3)).
(3)$$\mathrm{SH}\;(\mu \mathrm{mol}/g)=\frac{73.53\;\times \;A\;\times \;D}{C}$$
where A is the absorbance at 412 nm, C represents the concentration of freeze-dried dough in mg/mL, and D is the dilution factor of supernatant.

### 2.9. Cooking Properties and Texture Analysis of FCNs

Referred to the AACC 6650 (AACC, 2000), FCNs (30 g/500 mL water) were recooked for 90 s and rinsed in ice-cold water for 1 min. The cooking loss (CL) was defined as the percentage of dry matter that the noodles lose during cooking. The water absorption (WS), which was expressed as the percentage of water absorbed by noodles during cooking, was calculated as the mass ratio after and before cooking.

The texture profile analysis and tensile tests of recooked FCNs were carried out referring to our previous method [12], and measured within 10 min after recooking.

### 2.10. Organoleptic Evaluation

The recooked FCNs were evaluated by 30 trained panelists (male: female = 1:1) from the College of Food Science, Northeast Agricultural University. The samples were cut into 5 cm pieces and then cooked for 90 s in boiling water at a ratio of 1:10. Then, the samples were placed on dishes marked with random three-digit numbers and provided to all panelists in random order. Some organoleptic attributes such as appearance, texture, flavor, taste, and overall acceptability of FCNs were assessed. A nine-point hedonic scale ranging from 1 (dislike extremely) to 9 (like extremely) was used [19].

### 2.11. Statistical Analysis

All determinations were performed at least in triplicates. The data were expressed as the mean ± standard deviation (SD) and were examined by one-way analysis of variant (ANOVA) using SPSS (Version 13.0 for Windows, SPSS Inc., Chicago, IL, USA). Differences were considered to be significant at *p* < 0.05.

## 3. Results

### 3.1. Rheological Properties Analysis

#### 3.1.1. Dynamic Frequency Sweep

As depicted in Figure 1A,B, the G′ and G′′ of the dough containing NaCl or kansui were higher than those of the control, indicating the degree of gluten cross-linking and the hydration capacity of the dough was enhanced [20]. From Figure 1C, the tan δ was less than 1 for all dough samples, suggesting that the elastic behavior dominated over the viscous component in the entire frequency range investigated. The tan δ of the doughs containing 1% kansui was lower than that of others obviously, indicating that the elastic properties of the dough and the polymerization of gluten proteins were dramatically enhanced [21]. Moreover, the tan δ of the dough containing 3% NaCl was higher than that of the dough containing 1% and 2% NaCl significantly, which might be due to the higher levels of NaCl induced conformational changes and enhanced hydrogen bonding between the gluten components. The viscous flow of glutenin molecules relative to each other with gliadin as the molecular bearing during small strain measurements was conferred, resulting in a higher tan δ and a more viscoelastic network [22].

As listed in Table 1, the Power-law model is well fitted to the rheological property curves of each sample, with the corresponding coefficients of determination (*R*^2^) ranging from 0.952 to 0.995. The z′ values of all samples ranged from 0.244 to 0.261, indicating that the gluten network structure was not stable and the covalent bond structure had a certain tendency of physical interaction [15]. Furthermore, the control dough had the lowest K value (0.992 × 10^5^), which demonstrated the lower strength of wheat dough [23]. Dough with 1% kansui had the highest K value (2.118 × 10^5^), followed by that of the 1.5% kansui, 2% NaCl, and 0.5% kansui addition. This demonstrated that the addition of kansui to the dough could enhance gluten strength and noodle texture markedly, especially for doughs with 1% kansui.

#### 3.1.2. Creep and Recovery Measurements

Figure 1D exhibited the plots of strain as a function of time for dough samples. The addition of NaCl/kansui decreased the J_max_ of the dough significantly (*p* < 0.05) (Table 1), suggesting that NaCl/kansui enhanced the deformation resistance and rigidity of the dough, especially at 1% kansui addition [15]. The η_0_ showed a similar trend with a reduction of at least 27.82% and 52.69%, respectively. During the creep phase, η_0_ indicated the flowability of dough, which suggested that the addition of NaCl or kansui made the dough less fluid at the end of the applied load, the dough was less soluble in the solvent and was more stable to maintain a better-processed shape [24]. With increasing NaCl, the minimum value of η_0_ (8.74 × 10^9^ Pa·s) was achieved for doughs containing 2% NaCl, showing that this dough had a stronger structure and was more conducive to their subsequent recovery.

In the recovery phase, supplementation of NaCl or kansui induced higher J_e_/J_max_ and lower J_v_/J_max_, which suggested a higher dough recovery rate and the formation of a denser cross-linked gluten network. The dough containing 1% kansui exhibited the maximum J_e_/J_max_ (78.27%) and the maximum J_v_/J_max_ (21.73%), which further confirmed that this group presented a more desirable network structure and greater stability against dough film rupture [25].

### 3.2. Morphological Characteristics

As revealed in Figure 2, many voids were observed for the control, which might be related to the sublimation of the ice crystals forming voids in the noodles, while a looser gluten network also caused starch granules to fall off, thus creating voids [25,26]. With the addition of NaCl and kansui, more starch granules were embedded well in the cross-section and the gluten network was more continuous, suggesting that the strengthened gluten network could capture the starch firmly. As NaCl levels increased from 1% to 2%, the gluten network changed from less-connected protein particles to an elongated fibrous protein network with stronger adhesion of starch granules. When NaCl was added over 2%, a few loosely rounded starch granules surrounded by a less dense protein network was observed. This might be explained by the fact that at low levels (<2%), NaCl shielded the charges of the gluten protein, which promoted the slow hydration of the flour and created a more orderly network. Nevertheless, at high levels of NaCl (>2%), excessive shielding and repulsion occurred, with the NaCl preferring to bind to water, a fact that is detrimental to the hydration of gluten proteins [8,13].

Similar phenomena were observed in kansui treatment group, with kansui increasing, the number of pores decreased and induced denser membrane-like gluten protein structures. Especially with 1% kansui level, the starch granules were embedded better in the continuous protein network than other samples. However, excessive kansui resulted in deformed membrane-like gluten network structures with surface swollen starch granules that were not properly embedded in the protein network and could leach into the water during cooking, deteriorating the cooking quality of FCNs. Meanwhile, the porous structure of the unembedded starch granules might also produce more solids loss when the kansui was insufficient [4].

### 3.3. Freezable Water Content

As illustrated in Figure 3A–C, both NaCl or kansui led to a significant decrease of ΔH and FW content of FCNs (*p* < 0.05). This indicated that the ice crystallization ability and the number of ice crystals in FCNs decreased and the addition of NaCl/kansui could effectively inhibit the progressive liberation of water from the gluten network and ice recrystallization in the dough. These results might be related to the existence of non-covalent interactions between ionic compounds and water, which improved hydration capacity of the dough [27]. Furthermore, the appropriate concentration of NaCl/kansui could reduce the water migration of frozen dough by inducing a denser gluten structure, thus reducing the number and volume of crystals. The minimum values of ΔH (131.77) and FW (57.53%) were achieved at the 1% kansui level, indicating greatly enhanced water holding capacity of alkaline noodles. The probable reason was that NaCl prompted a slight depolymerization of proteins during cooking, while kansui contributed to protein aggregation [11,28], thereby forming a higher spatial barrier. This might also be related to the kansui contributing to an increase of the hydrophilic phase viscosity, resulting in less movement of water molecules that could form ice crystals [29].

### 3.4. Water Distribution and Migration

#### 3.4.1. Water Distribution

The *T*_2_ spectra as shown in Figure 3D was inverted from the attenuation curve obtained by LF-NMR technology. Three ^1^H *T*_2_ populations were observed and named starting from the shorter to the longer relaxation time *T*_21_ (0.1–1 ms), *T*_22_ (1–10 ms), and *T*_23_ (10–100 ms), and their proportions were named *A*_21_, *A*_22_, and *A*_23_, respectively. The relatively short *T*_21_ represented the tight binding of bound water to non-aqueous components, such as a gluten matrix consisting mainly of protein and starch. The *T*_22_ composition denoted semi-bound water. In this state, the combination of water molecules with gluten and starch was relatively loose, filling mainly the space of the parallel sheet in gluten. The comparatively long *T*_23_ was designated as free water, which was mainly distributed in the gluten network voids and was the most mobile [30].

The addition of NaCl/kansui reduced the *T*_23_ values of FCNs significantly (*p* < 0.05), indicating that the interaction between water and non-aqueous components was enhanced and the compatibility was improved. Interestingly, the peak apexes of the three populations were shifted to the left distinctly by adding kansui, with *T*_21_ and *T*_22_ also decreased from 0.28 ms and 2.77 ms to 0.21 ms and 2.11 ms, respectively, indicating that kansui not only induced greater binding effects between aqueous and non-aqueous components, but also encouraged the dough to exhibit stronger spatial hindrance to delay deeply absorbed water desorbing (as shown in Figure 3E,F). These findings were further confirmed by the *A*_21_ (4.54–6.77%) and *A*_22_ (2.48–3.15%) increased significantly and the *A*_23_ (92.99–90.08%) of the alkaline noodles decreased correspondingly (*p* < 0.05).

With increasing levels of kansui, the relaxation time showed insignificant changes, while the FCNs containing 1% kansui achieved maximum values of *A*_21_ (6.89%) and *A*_22_ (3.87%) as well as the minimum values of *A*_23_ (89.25%). The sharp drop in free water content of alkaline noodles meant that kansui could reduce the loss of bound water in the dough system more effectively. This might be related to the proper levels of kansui could induce protein polymerization through disulfide bonds or hydrophobic interactions, and form a more compact network structure [9]. Moreover, the appropriate levels of kansui helped to maintain the stability of starch granules, which prevented them from being exposed to hydrophobic groups due to damage, thereby contributing to the high hydration capacity of starch during freezing [12]. 

With increasing NaCl, the minimum values of *T*_23_ (36.12 ms) and *A*_23_ (91.92%) related to free water were both obtained at 2% NaCl level. Na^+^ ions could prevent starch retrogradation during aging, making starches less likely to build up and loosen, and prompting faster water migration through noodles. However, due to the formation of the protein network, the fewer pores on the surface of FCNs with NaCl, which decreased the channels for water to penetrate into the noodle [8]. Therefore, 2% NaCl induced the formation of a denser gluten network, and the effect of protein network formation on the water migration speed was much greater than the effect of reduced starch lumps size. Overall, water migration could be limited by the addition of appropriate levels of NaCl/kansui, which was consistent with the DSC results.

#### 3.4.2. Water Migration

As illustrated in Figure 4, the addition of NaCl and kansui could induce lower water mobility and excellent water retention in FCNs. As NaCl and kansui were increasing, the red areas occupied most of the cross section gradually and the green areas were distributed evenly, indicating that the addition of NaCl and kansui had an inhibitory effect on water migration and dehydration. Especially for the 2% NaCl and 1% kansui groups, a well-integrated circle of high hydration might be noticed on the outside and the water penetrated gradually from the surface to the interior of the cooked noodles. This suggested that suitable additions of NaCl and kansui could promote the formation of a coherent gluten network [31]. The red area of Figure 4 (K-1) appeared to be higher than that of (N-2), showing that 1% kansui provided the dough with the best water retention performance. Furthermore, alkaline noodles exhibited a large difference in water content between the surface and the center, with a higher moisture gradient, which could contribute to a firm texture of FCNs.

### 3.5. Gluten Secondary Structure

As described in Figure 5A, both NaCl and kansui added to the dough caused a significant increase in β-sheets and β-turns, while α-helixes and random coils decreased significantly (*p* < 0.05). Generally, β-sheet is recognized as the most stable secondary structure among all of them, which itself increased molecular rigidity and frozen dough strength [28]. As for β-turns, Bock and Damodaran (2013) reported that β-turns were the preferred structure of gluten in the fully hydrated state [32]. Furthermore, consecutive β-turns could form β-spirals domains with short α-helixes on the flank, and the β-spirals structures contributed to the viscoelasticity of dough [33], which was in line with the rheological results (Section 3.3). Lower contents of α-helixes, indicating a change in the order of hydrogen bond and a redistribution of water in frozen doughs. The random coil was typically regarded as a disordered secondary structure, and its reduction suggested that the addition of NaCl/kansui contributed to the orderliness of protein secondary structure.

Figure 5B displayed the changes in the ordered and unordered portions of the protein. The ordered structure consisted of the β-sheets and α-helixes portions of protein, while the unordered structure involved β-turns and random coils [34]. The control had 57.92% ordered structure and 42.08% unordered structure. The addition of 1% kansui increased the ordered structure of doughs by 12.22%, with a corresponding decrease in the unordered structure by 19.13%. This could mainly be attributed to their higher content of α-helixes (33.73%) and β-sheets (32.25%) conformation, which contributed to inducing stronger intramolecular hydrogen bonds and leading to a more compact and stable dough structure [14]. Based on the stable and tough nature of the α-helix structure, its increased content also played an important role in improving the hardness of dough [35]. Furthermore, according to Belton et al. (1995), β-sheets depended on the hydration of glutenin [36]. The higher β-sheets further confirmed the contribution of kansui to the water retention capacity of gluten in FCNs. These results might suggest that appropriate levels of kansui induced greater unfolding and aggregation of polypeptides than NaCl, leading to more order and dense gluten protein network. 

### 3.6. Changes of Free Sulfhydryl (-SH) Contents

From Figure 5C, disulfide bonds are essential for maintaining the gluten proteins structure during freezing, and their integrity is reflected in the content of free -SH. The addition of NaCl or kansui reduced the -SH content of doughs significantly (*p* < 0.05) compared with the control. With increasing NaCl, the minimum free sulfhydryl content (6.19 μmol/g) of FCNs was reached at the level of 2% NaCl. At 1% and 3% concentrations of NaCl, the interactions between the gluten proteins are too weak or strong to aggregate in a rigid form, making the reorientation of the sulfhydryl group contacts difficult and the dough had a higher content of -SH. However, 2% NaCl caused local fluctuations in the relative orientation of the glutenins, contributing to the sulfhydryl groups contacts and providing for their subsequent cross-linking, which was fundamental to the glutenin network formation [8]. Overall, the -SH content of the frozen dough showed a wavy trend, with a minimum value (5.53 μmol/g) achieved at the 1% kansui level. Theoretically, the free -SH underwent two chemical changes in wheat gluten with kansui increasing (Figure 5D), (i) oxidation of the free -SH and SH-SS interchange reactions consume part of the free -SH groups and produce SS cross-linking, and (ii) the β-elimination reaction, which is more likely to occur under high alkali conditions, causes a significant increase in -SH content, which fitted well with our results [5,12].

### 3.7. Quality Characteristics

#### 3.7.1. Cooking Properties

As summarized in Table 2, the WS and CL of noodles varied from 132.95% to 135.28% and 4.77% to 4.89% with increasing NaCl, respectively, which were slightly higher than those of the control (130.73% and 4.49%), but these variations were not significant (*p* > 0.05). However, kansui increased the WS and CL of recooked FCNs significantly, and with kansui increasing, the WS (146.35–155.87%) and CL (5.92–7.88%) increased significantly (*p* < 0.05). The higher WS induced by kansui was related to their stronger penetration effect, which might lead to a faster water absorption capacity of doughs during mixing and sheeting, thereby developing a stronger gluten network [8]. Moreover, the CL of noodles with 1.5% kansui approached twice that of the control. This may be because kansui promoted faster heat transfer by raising the boiling point of the cooking water, resulting in more soluble substances (such as some α- and γ-gliadins) dissolving into the cooking water and thus increasing the solids loss [28].

#### 3.7.2. Texture Properties

The NaCl and kansui played different roles in alleviating the decline in hardness and tensile properties of FCNs (Table 2), NaCl resulted in superior dough extensibility, such as tensile strength and breaking distance increased by at least 20.62% and 27.49%, respectively, while kansui increased noodle hardness (52.74–64.00 N) and chewiness (183.35–282.48) more significantly. The maximum values of tensile strength (90.35 g) and breaking distance (61.94 mm) were reached at 2% NaCl addition, while higher levels resulted in a significant decrease in the tensile properties of the FCNs (*p* < 0.05). This might be due to the appropriate levels (2%) of NaCl could shield the charges on the proteins and contribute to the elongation of the gluten network [20]. However, the lower tensile capacity of FCNs at higher concentrations of NaCl might be related to the excessive water intake of noodles during the steaming process, yielding noodles with softened and less recoverable texture as more starch granules are swelled.

With increasing kansui levels, the noodle hardness and chewiness, which were related to gluten strength, showed a concentration-dependent upward trend, while no significant differences were observed after the 1% kansui level (*p* > 0.05). Kansui could form a tighter and more rigid network structure by promoting protein cross-linking, which might limit the collapse of starch molecules and increase the dough strength [5]. Notably, the tensile strength (78.79–72.56 g) and breaking distance (52.32–48.56 mm) of the FCNs decreased significantly (*p* < 0.05) as the kansui levels increased from 1% to 1.5%, which could be explained as the gluten network in alkaline noodles was already too strong or too firm to maintain sufficient springiness to cushion the expansion of the starch granules produced during cooking before the starch granules were fully pasted, resulting in a less tensile capacity of FCNs [28].

### 3.8. Organoleptic Properties

As shown in Figure 6, as the levels of NaCl and kansui increased, the improvement effect on the organoleptic properties of noodles became more significant. In particular, noodles with 1% kansui had the best sensory scores in terms of appearance, taste, flavor, texture, and overall acceptability, indicating that consumers preferred noodles with a firm texture and high chewiness. The cooked noodles with kansui in the formulation had enhanced aroma compared to the NaCl, indicating that kansui could impart a more desirable flavor to FCNs. This might be related to the fact that kansui could inhibit the formation of non-aroma flavor compounds caused by fat oxidation by adjusting the pH of the dough [37]. Moreover, the addition of 2% NaCl improved the appearance, taste, texture, and overall acceptability of the noodles markedly, which could be explained by the appropriate levels of NaCl induced the superiority in dough extensibility and recovery ability.

### 3.9. Network Reinforcement Mechanism by NaCl and Kansui after Freezing

A model to explain the mechanism whereby NaCl and kansui affect gluten proteins network formation after freezing was proposed in Figure 7. Previous studies confirmed that frozen storage mainly led to the disaggregation of protein aggregates by destroying the disulfide cross-links between gluten proteins [38]. In our study, 1% kansui induced more stable intermolecular disulfide bonding in the dough during freezing, which contributed to the more complete gluten structure and stronger water retention capacity of FCNs. Kansui could induce stronger hydrophobic interactions between gluten proteins compared to NaCl has also been confirmed [9]. In summary, NaCl and kansui exerted different effects on the stability of the gluten network structure of FCNs after freezing. Kansui stimulated the conversion of free sulfhydryl groups to disulfide bonds obviously, generating denser crosslinks based on the original network. Thus, a greater mechanical force was required to break the alkaline dough. In contrast, NaCl tended to enhance the original gluten network structure, but contributed to the extensibility of the frozen dough, which could also explain the difference in quality of the recooked FCNs.

## 4. Conclusions

There is no doubt that the quality of frozen cooked noodles with NaCl or kansui showed a significant difference, as explained by the variation in rheological behavior, microscopic morphology, water distribution, and protein structure of dough. Both the addition of NaCl and kansui to the dough formulation improved the organoleptic properties of FCNs. NaCl increased the tensile strength and breaking distance of FCNs by at least 20.62% and 27.49%, respectively; the 2% NaCl level especially had the best tensile properties of FCNs, which corresponded to the more elongated fibrous protein network. NaCl had no significant effect on the cooking quality of frozen cooked noodles. However, kansui increased the water absorption of recooked FCNs significantly (*p* < 0.05), corresponding to the lower freezable water content and higher binding capacity for deeply adsorbed water. The maximum hardness (65.55 N) and chewiness (296.04) were reached at 1% kansui levels, which was consistent with the denser membrane-like protein structures, the strongest intermolecular disulfide bonds and the highest protein ordered secondary structure. Rheological results further confirmed that the resistance to deformation and recovery ability of thawed dough were improved significantly at 1% kansui level. The experimental results obtained in this study may give the noodles industry more fundamental insight into the effect of NaCl or kansui on the quality attributes of frozen cooked noodles.

## Figures and Tables

**Figure 1 foods-10-03132-f001:**
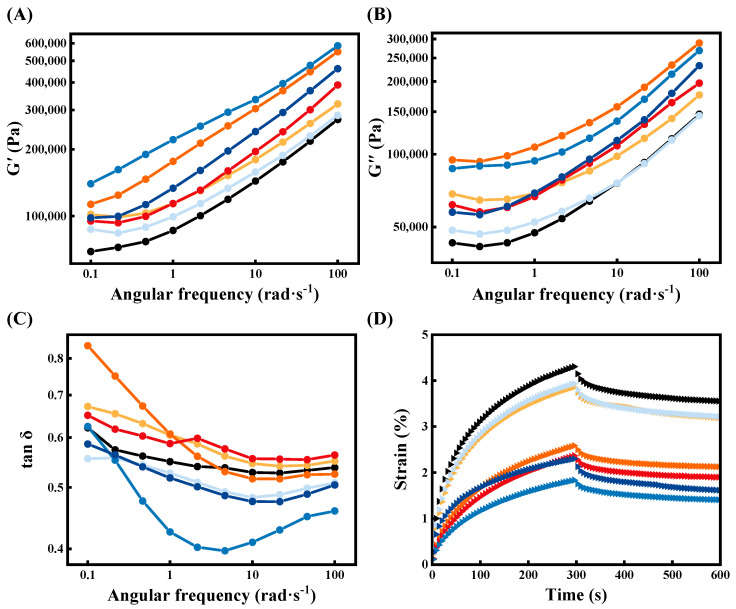
Mechanical spectra and creep-recovery curves of frozen dough. (**A**) G′, (**B**) G′′, and (**C**) tan δ versus frequency. (**D**) The creep-recovery curves. 

 Control, 

 N-1, 

 N-2, 

 N-3, 

 K-0.5, 

 K-1, 

 K-1.5.

**Figure 2 foods-10-03132-f002:**
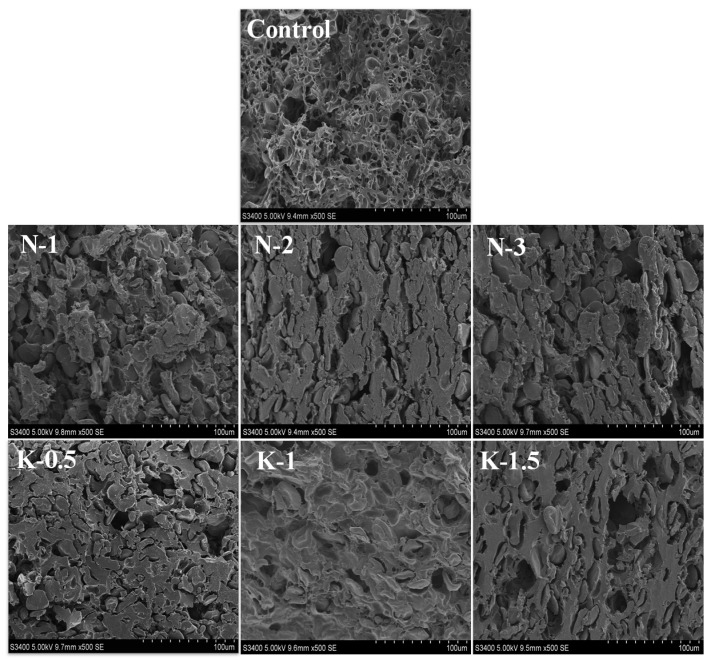
Microscopic morphology of lyophilized frozen cooked noodles.

**Figure 3 foods-10-03132-f003:**
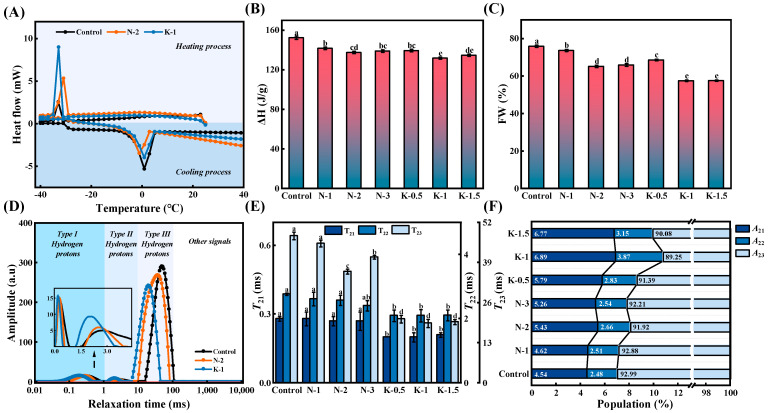
Water distribution of the frozen cooked noodles. (**A**) Freezing and melting curves. The (**B**) melting enthalpy (ΔH) and (**C**) freezable water content (FW) of frozen cooked noodles. (**D**) The typical *T*_2_ relaxation time distribution curve. The (**E**) relaxation times and (**F**) peak area proportions of frozen cooked noodles. Different letters in the same bar chart represent significant differences (*p* < 0.05).

**Figure 4 foods-10-03132-f004:**
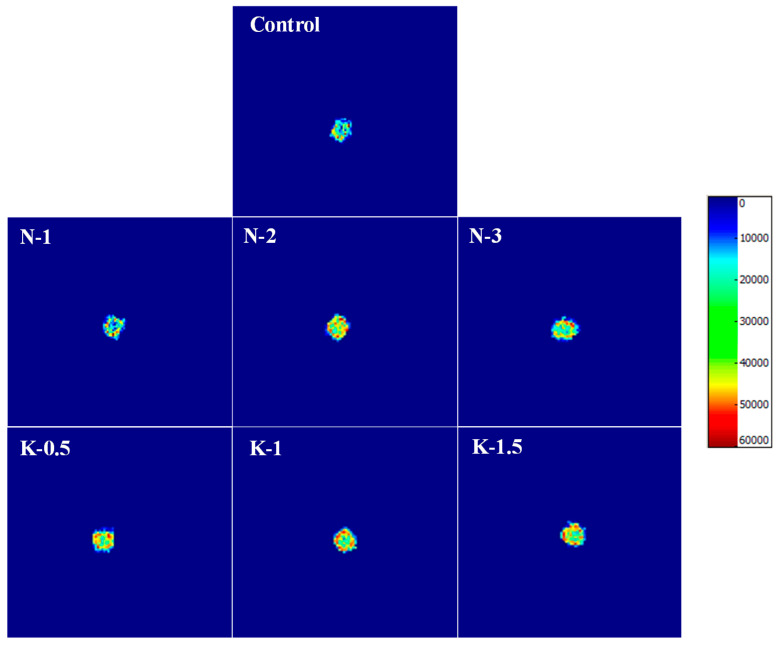
MRI images of the frozen cooked noodles.

**Figure 5 foods-10-03132-f005:**
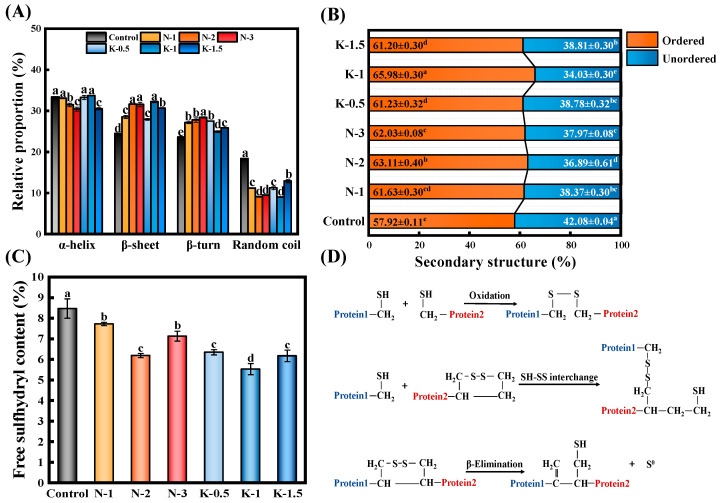
Structural properties of gluten protein. (**A**) Relative proportion of secondary structure. (**B**) Breakdown of ordered and unordered structures in gluten protein. (**C**) The free sulfhydryl content of frozen dough. (**D**) Overview of some common reactions in or between amino acid chains which are enhanced by alkaline conditions. Different letters in the same bar chart represent significant differences (*p* < 0.05).

**Figure 6 foods-10-03132-f006:**
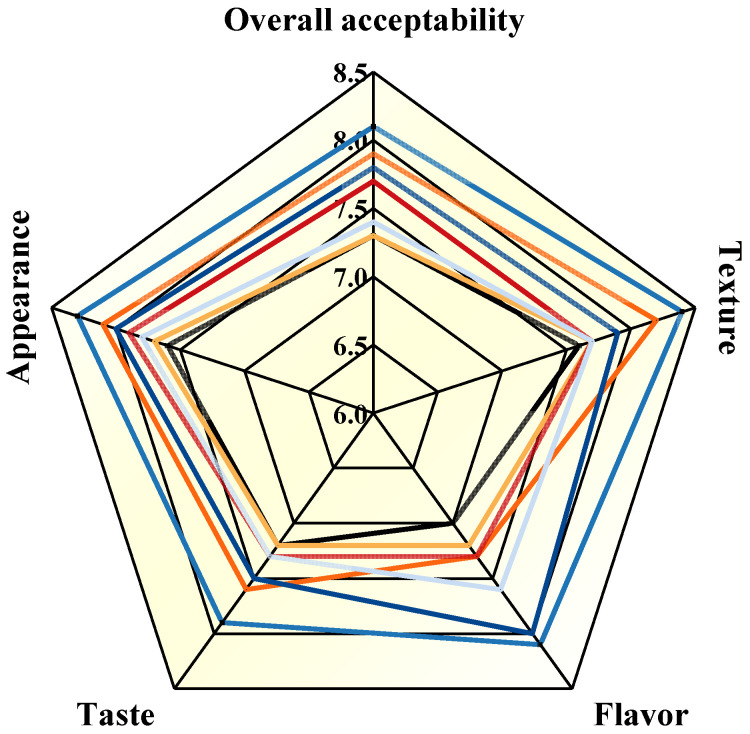
Organoleptic evaluation parameters of frozen cooked noodles. 

 Control, 

 N-1, 

 N-2, 

 N-3, 

 K-0.5, 

 K-1, 

 K-1.5.

**Figure 7 foods-10-03132-f007:**
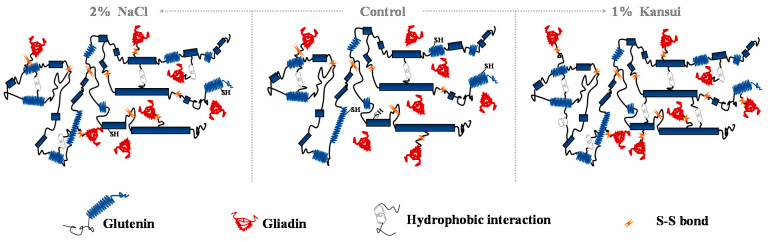
The proposed mechanism for NaCl and kansui to reinforce the gluten networks in frozen dough.

**Table 1 foods-10-03132-t001:** The key parameters of the power law model and creep recovery curve of frozen dough with NaCl or kansui.

Samples	z′	K × 10^5^	R^2^	Creep Phase		Recovery Phase	
				J_max_ × 10^−4^ (1/Pa)	ƞ_0_ × 10^9^ (Pa·s)	J_e_/J_max_ (%)	J_v_/J_max_ (%)
Control	0.261 ± 0.017 ^a^	0.992 ± 0.037 ^e^	0.952 ± 0.049	1.24 ± 0.09 ^a^	15.96 ± 0.92 ^a^	54.75 ± 3.68 ^d^	45.26 ± 3.68 ^a^
N-1	0.246 ± 0.052 ^a^	1.055 ± 0.049 ^de^	0.995 ± 0.002	1.09 ± 0.11 ^bc^	11.52 ± 1.12 ^b^	63.15 ± 2.03 ^c^	36.85 ± 2.03 ^b^
N-2	0.246 ± 0.015 ^a^	1.184 ± 0.062 ^bc^	0.968 ± 0.046	0.97 ± 0.07 ^cd^	8.74 ± 0.77 ^c^	73.57 ± 1.34 ^b^	26.43 ± 1.34 ^c^
N-3	0.257 ± 0.007 ^a^	1.071 ± 0.025 ^de^	0.977 ± 0.001	0.95 ± 0.05 ^d^	9.33 ± 0.63 ^c^	72.72 ± 3.06 ^b^	27.28 ± 3.06 ^c^
K-0.5	0.254 ± 0.003 ^a^	1.114 ± 0.073 ^cd^	0.982 ± 0.010	1.11 ± 0.04 ^b^	7.55 ± 0.25 ^d^	65.95 ± 3.17 ^c^	34.05 ± 3.17 ^b^
K-1	0.244 ± 0.004 ^a^	2.118 ± 0.031 ^a^	0.993 ± 0.004	0.77 ± 0.03 ^e^	6.92 ± 0.35 ^d^	78.27 ± 1.22 ^a^	21.73 ± 1.22 ^d^
K-1.5	0.251 ± 0.006 ^a^	1.254 ± 0.077 ^b^	0.980 ± 0.009	0.93 ± 0.09 ^d^	7.34 ± 0.50 ^d^	74.23 ± 0.72 ^b^	25.77 ± 0.72 ^c^

z′, the degree of dependence of G′ on frequency sweep. K, the strength of molecular interactions. *R*^2^, the corresponding coefficients of determination. J_max_, maximum creep compliance. η_0_, the zero-shear viscosity. J_e_/J_max_, the relative elastic part of maximum creep compliance. J_v_/J_max_, the relative viscous part of maximum creep compliance. Different superscripts in the same column indicate significant difference (*p* < 0.05).

**Table 2 foods-10-03132-t002:** Quality characteristics of frozen cooked noodles with NaCl or kansui.

Samples	Cooking Properties		Textural Properties			
	WS (%)	CL (%)	Hardness (N)	Chewiness	Tensile Strength (g)	Breaking Distance (mm)
Control	130.73 ± 5.40 ^c^	4.49 ± 0.16 ^c^	52.74 ± 3.81 ^c^	183.35 ± 15.96 ^c^	66.82 ± 2.64 ^e^	43.03 ± 5.00 ^d^
N-1	132.95 ± 3.22 ^c^	4.77 ± 0.07 ^c^	55.89 ± 3.57 ^c^	208.70 ± 11.94 ^c^	80.60 ± 1.18 ^bc^	57.18 ± 2.19 ^ab^
N-2	133.37 ± 2.73 ^c^	4.82 ± 0.22 ^c^	56.94 ± 1.59 ^c^	216.81 ± 19.63 ^bc^	90.35 ± 2.81 ^a^	61.94 ± 2.75 ^a^
N-3	135.28 ± 2.57 ^c^	4.89 ± 0.36 ^c^	56.14 ± 2.52 ^c^	211.89 ± 17.85 ^bc^	82.96 ± 2.87 ^b^	54.86 ± 3.03 ^b^
K-0.5	146.35 ± 1.24 ^b^	5.92 ± 0.24 ^b^	58.89 ± 4.26 ^bc^	246.83 ± 21.87 ^b^	74.88 ± 1.58 ^cd^	46.45 ± 1.52 ^d^
K-1	146.42 ± 1.75 ^b^	6.35 ± 0.24 ^b^	65.55 ± 3.83 ^a^	296.04 ± 28.52 ^a^	78.79 ± 6.13 ^bcd^	52.32 ± 3.91 ^bc^
K-1.5	155.87 ± 5.36 ^a^	7.88 ± 0.09 ^a^	64.00 ± 3.62 ^ab^	282.48 ± 23.47 ^a^	72.56 ± 4.89 ^de^	48.56 ± 2.37 ^cd^

WS, water absorption. CL, cooking loss. Different superscripts in the same column indicate significant difference (*p* < 0.05).

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
