# Peer review of "Moisture Distribution and Structural Properties of Frozen Cooked Noodles with NaCl and Kansui"

_foods, 2021, doi:10.3390/foods10123132_

Round 1
Reviewer 1 Report
The article titled "Moisture Distribution and Structural Properties of Frozen Cooked Noodles with Salt and Kansui" deals with an interesting aspect of the influence of the preparation method of noodles on their quality after freezer processing. The advantage of the study is the large number of instrumental analyses and their detailed interpretation of the results carried out by the authors. My comment concerns the lack of organoleptic evaluation. It seems that organoleptic evaluation allows differences in quality that are impossible to capture by other methods. A less important comment concerns the more detailed description of the freezing and thawing method. The rate of freezing/thawing can have a significant effect on the amount of water frozen and the size of the ice crystals formed e.g. it is not clear from the text whether the freezing was done in air under free convection or perhaps immersion and whether the air was removed from the foil pouch and whether it was thawed in pouches.
Author Response
Response to Reviewer 1 Comments
Manuscript ID: foods-1482336
Title: Moisture Distribution and Structural Properties of Frozen Cooked Noodles with Salt and Kansui
Your comments were highly insightful and enabled us to greatly improve the quality of our manuscript. In accordance with your comments, we have had the manuscript professionally edited, explained the rationale for our study in more detail. In the following pages are our point-by-point responses to each of the comments. We hope that the revisions in the manuscript and our accompanying responses will be sufficient to make our manuscript suitable for publication in Foods. Comments 1. My comment concerns the lack of organoleptic evaluation. It seems that organoleptic evaluation allows differences in quality that are impossible to capture by other methods. Response: According to your perfect advice, we have carried out the organoleptic evaluation experiments and added detailed information to the manuscript.
Line 321-328 (Page 4):
2.10. Organoleptic evaluation
The recooked FCNs were evaluated by 30 trained panelists (male: female= 1:1) from the College of Food Science, Northeast Agricultural University. The samples were cut into 5 cm pieces and then cooked for 90 s in boiling water at a ratio of 1:10. Then, the samples were placed on dishes marked with random three-digit numbers and provided to all panelists in random order. Some organoleptic attributes such as appearance, texture, flavor, taste, and overall acceptability of FCNs were assessed. A nine-point hedonic scale ranging from 1 (dislike extremely) to 9 (like extremely) was used (Sofi, Singh, Chhikara, Panghal, & Gat, 2020).
Line 750-763 (Page 13):
3.8. Organoleptic properties
Figure 6. Organoleptic evaluation parameters of frozen cooked noodles. Control, N-1, N-2, N-3, K-0.5, K-1, K-1.5.
As shown in Figure 6, as the levels of salt and kansui increased, the improvement effect on the organoleptic properties of noodles became more significant. In particular, noodles with 1% kansui had the best sensory scores in terms of appearance, taste, flavor, texture, and overall ac-ceptability, indicating that consumers preferred noodles with a firm texture and high chewiness. The cooked noodles with kansui in the formulation had enhanced aroma compared to the NaCl, indicating that kansui could impart a more desirable flavor to FCNs. This might be related to the fact that kansui could inhibit the formation of non-aroma flavor compounds caused by fat oxida-tion by adjusting the pH of the dough (Xi, Xu, Wu, Jin, & Xu, 2020). Moreover, the addition of 2% NaCl improved the appearance, taste, texture, and overall acceptability of the noodles markedly, which could be ex-plained by the appropriate levels of salt induced the superiority in dough extensibility and re-covery ability.
Line 796-797 (Page 14):
Both the addition of salt and kansui to the dough formulation improved the organoleptic properties of FCNs.
Sofi, S. A., Singh, J., Chhikara, N., Panghal, A., & Gat, Y. (2020). Quality characterization of gluten free noodles enriched with chickpea protein isolate. Food Bioscience, 36, 100626.
Xi, J., Xu, D., Wu, F., Jin, Z., & Xu, X. (2020). Effect of Na2CO3 on quality and volatile compounds of steamed bread fermented with yeast or sourdough. Food Chemistry, 324, 126786.
Comments 2.
A less important comment concerns the more detailed description of the freezing and thawing method. The rate of freezing/thawing can have a significant effect on the amount of water frozen and the size of the ice crystals formed e.g., it is not clear from the text whether the freezing was done in air under free convection or perhaps immersion and whether the air was removed from the foil pouch and whether it was thawed in pouches.
Response: Sorry, it was our mistake. To make it more precise, we have revised relevant description in our paper.
Lines 99-105 (Page 2):
These noodles were drained for 1 min and placed in sealed bags. Then, they were frozen in an ultra-low temperature freezer (DW-HL100, MELNG, Hefei, China), with air temperature in convection at -40 °C, until the core temperature dropped to -18 °C (~60 min). After freezing, the FCNs were packaged in polyethylene bags and stored at -18 ± 2 °C in the freezer (BCD-480WDGB, Haier Co., Ltd., Qingdao, China) for more than 24 h until use. Also, some fresh dough sheets were cut into 35 mm diameter discs and placed in polyethylene bags and frozen in the same procedure for further rheology testing.
Lines 107-109 (Page 2):
After fixed-time frozen storage, the dough sheets were removed from the polyethylene bags and put in the stainless steel tray. Then, they were thawed in a constant temperature incubator at 25 °C and 75% relative humidity for 1 h.
We sincerely hope that our work and our revise make you satisfied. Thank you once again.

Reviewer 2 Report
The manuscript is very well written and the subject matter is appropriate for Foods and interesting for the readers. Numerous characterization techniques have been used and the materials and methods used are appropriate for the research carried out. The results are well analyzed and the conclusions are logical and well founded. The main criticisms that I can make to this work are the presentation of the results. I believe that some aspects should be improved and certain questions answered.
- Figures 1A to 1C. The name of this plots are mechanical spectra, not rheological properties. A log-log scale must be used. In addition, 1B can be incorportaed in 1A.
- I assume that stress sweeps were carried out to determine the critical strain. However, if the rheometer is stress controlled, I don't know why critical stress was not used as a criterion, it is the most logical thing to do.
- Why have no flow curves been made?
- Figures 1E. It is absolutely impossible to appreciate anything, not even scale, with that size of figure. I suggest putting it as a separate and larger figure.
- In table 1, the parameter K depends on its units of z´. From a mathematical point of view it is not correct, although physically it may make sense. I suggest modifying equation 1 as, for example, is done in the following script with the flow curves:
- Figura 2G. Same as Fig 1E.
